# OpenReview forum: "Video-MTR: Reinforced Multi-Turn Reasoning for Long Video Understanding"
_ICML.cc/2026/Conference — ICML 2026 regular_

### Official Review · Reviewer_MMsX · 2026-03-10

**Soundness:** 2
**Presentation:** 2
**Significance:** 2
**Originality:** 4
**Overall Recommendation:** 4
**Confidence:** 4

**Summary:**

This paper proposes Video-MTR, a framework that reformulates long-form video understanding as a multi-turn interactive reasoning task. An MLLM agent iteratively selects and retrieves key video segments based on the question and previous visual context. The authors introduce a gated bi-level reward system for pure RL post-training, which rewards intermediate frame-retrieval relevance and final answer correctness. The model demonstrates significant data efficiency, achieving good results on video benchmarks.

**Compliance With Llm Reviewing Policy:**

Affirmed.

**Final Justification:**

The limitation is still there and not easy to resolve. So I will keep my score

**Key Questions For Authors:**

1. In Table 5, accuracy gains and interaction turns appear to diminish beyond 3 turns while latency continues to rise. Could you explain this?

**Limitations:**

yes

**Strengths And Weaknesses:**

**Strengths**
1. The shift from static, single-turn sampling to a dynamic, interactive loop is an excellent idea! And the technical contribution in multi-turn video training is significant.
2. The framework shows strong data efficiency to achieve SOTA performance.
3. The introduction of a goal-gated mechanism, where turn-level rewards are only granted if the final answer is correct, is a clever and novel solution.


**Weaknesses**
1. Inference and evaluation is not clear.
   - The model is capped at 80 frames , which is substantially lower than the 256+ frames used by many baselines for hour-long videos. It is unclear if the performance gains hold when compared against these models at their native frame scales.
   - The comparison lacks a baseline where other open-source models (like the Qwen backbone) are given the same multi-turn inference protocol in a zero-shot setting to isolate the benefit of the RL training versus the prompting strategy.
   - It is not specified if the resolution was kept constant or if max_pixels was used in Qwen2.5-VL , which could lead to automatic resolution changes with different numbers of input frame.
2. There is a potential mismatch between the training data (likely shorter clips from NExT-GQA/QVHighlights) and the the test sets in the video duration. A common way to deal with long video understanding is to train the model on long videos.  Furthermore, the computational overhead of calculating the similarity reward for turn-level feedback may limit training scalability.
3. While the model is trained for "relevance," the paper does not quantitatively evaluate the hit rate: whether the model is truly identifying the most critical frames for complex reasoning.
4. Efficiency and scaling limitations:
   - The use of veRL and VAGEN for multi-turn RL is noted for its complexity , and the authors admit that training with hundreds of frames is currently computationally prohibitive.
   - As turn count increases, the inference latency grows nearly linearly (e.g., from 194ms at 1 turn to 622ms at 3 turns).
5. The work would benefit from a direct comparison to recent works in tool-use video RL or other multi-turn inference frameworks. Examples could be: https://arxiv.org/pdf/2509.21100, https://arxiv.org/pdf/2511.05489

Although I personally see the strong potential of the multi-turn interactive paradigm of the video RL training, the problems in the inference limit the significance of this work.

---

> ### Author Rebuttal · Authors · 2026-03-31
>
> Dear Reviewer MMsX,
>
> We sincerely thank you for the positive assessment and the constructive comments. We now respond to the specific questions listed above.
>
> **W1: Inference and evaluation is not clear.**
>
> 1. **Native-scale comparison is already included**. Our main table already compares against baselines at their native frame scales, many of which are much larger than ours. Specially, LongVA and Video-XL(256f+ ) both belong to the long-context optimization family, which process **more frames at larger frame scales.**
> 2. **Resolution is controlled in matched-budget comparisons**.  All matched-budget comparisons (32/64/80f) use the **same fixed** max_pixels = 128 × 28 × 28 for Qwen2.5-VL-7B, Video-MTR, and Video-R1.The 768-frame Qwen2.5-VL-7B number is quoted from the official report under its native setup.
> 3. **New results of zero-shot multi-turn setting**. We additionally evaluated a zero-shot multi-turn Qwen2.5-VL-7B baseline at 80 frames. It achieves  55.5%, 44.4%, 45.7%, 33.8%, 65.8%, versus 59.5%, 45.2%, 48.4%, 33.6%, 63.5% for the standard single-turn backbone on VideoMME, MLVU, LongVideoBench, LVBench, and EgoSchema. Thus, simply adding a multi-turn protocol without training does not help overall and in fact hurts most benchmarks. By contrast, Video-MTR outperforms both by a clear margin, indicating that the improvement comes from RL-trained multi-turn grounding and reasoning.
>
> **W2: Mismatch Between Training & Testing Data**
>
> We would like to clarify that our choice of NExT-GQA and QVHighlights is directly aligned with the goal of Video-MTR: to develop a **scalable and data-efficient** training paradigm without additional annotation. Concretely, we repurpose existing grounding benchmarks to provide temporal supervision at zero extra annotation cost. This is precisely what makes our training paradigm scalable: unlike contemporaneous methods such as TimeSuite ( $\sim$**432K** samples) or Temporal-RLT (**32K selected from a 490K** pool), Video-MTR achieves strong long-video performance with only $\sim$**8K** training examples. More importantly, this design naturally supports further scaling by incorporating future long-video grounding datasets into the same pipeline.
>
> **W3： “No quantitative evaluation of ‘hit rate’**
>
> We thank the reviewer for this important question. We would like to clarify that Video-MTR is not optimized for standalone temporal grounding, but for QA-conditioned evidence acquisition. In our framework, retrieval is only an **auxiliary signal** goal-gated by final answer correctness. Therefore, our goal is not to maximize grounding metrics, but to learn which evidence is actually useful for answering the question correctly.
>
> **W4: Efficiency and scaling limitations**
>
> We thank the reviewer for raising this point. The 80-frame cap reflects a practical post-training budget, not a limitation unique to Video-MTR: outside a few specialized long-context systems, most long-video post-training methods still operate at $\leq$ 128 frames. We believe larger-budget training should become feasible as more efficient multi-turn RL frameworks emerge.
> For inference latency, we fully agree that more turns increase cost. This is exactly why we provide a turn-limit analysis in Appendix A.3 and choose $K_{\max}=3$. Our goal is not to maximize turns, but to show that a small number of adaptive retrieval–reasoning steps already yields substantial gains.
>
> **W5:  Comparison to recent works in tool-use/multi-turn methods**
>
> We thank the reviewer for the suggestion. However, these methods operate at a fundamentally different scale, particularly in data volume and construction complexity. VideoChat-R1.5 leverages an additional **80K newly annotated, non-public training set**. Similarly, TIMESEARCH-R is trained on a large-scale, heavily curated long-video dataset, constructed from multiple sources (e.g., Haystack-Ego4D, Panda-70M, CinePile) and further refined through **multi-stage filtering, annotation via GPT-4o, and manual cleaning**.
>  In contrast, our approach does not rely on such scaling, yet still achieves competitive or superior results, suggesting that our gains arise from modeling and learning innovations.
>
> **Q1: Diminishing gains and turns beyond 3 turns**
>
> We believe the non-uniform gains and turn usage mainly arise from a train–test mismatch, not from a limitation of multi-turn reasoning itself. Current benchmarks increasingly emphasize minute-to-hour, highly challenging videos, while grounded training data at that scale remain limited. Under our **scalable, data-efficient** paradigm without additional long-video annotation, the policy sees limited supervision on the longest and hardest cases. The learned policy is therefore a practical balance given the current training corpus. This limitation is not intrinsic to Video-MTR: as larger temporally grounded long-video datasets become available, the same pipeline can naturally scale to higher frame and turn budgets.

---

> > ### Author Rebuttal · Reviewer_MMsX · 2026-04-04
> >
> > From the response, I still believe the main limitation of the framework is its scalability to more rounds, longer training data, or more frames, and this is not easy to resolve for now. I understand that the turn-wise retrieval is novel and efficient, but that does not mean we no longer need to scale it up. Given this, I will keep my score

---

> > > ### Author Response · Authors · 2026-04-06
> > >
> > > We thank the reviewer for the thoughtful feedback. We agree that scalability is an important direction. Our current design focuses on data-efficient multi-turn reasoning under practical constraints, and we view scaling as complementary future work. We expect this to further benefit from more publicly available grounded long-video data, and advances in efficient RL training frameworks. We appreciate your consideration.

---

### Official Review · Reviewer_tZbB · 2026-03-12

**Soundness:** 3
**Presentation:** 3
**Significance:** 2
**Originality:** 2
**Overall Recommendation:** 4
**Confidence:** 4

**Summary:**

In this paper, the authors propose Video-MTR, a multi-turn reasoning framework with RL post-training. Specifically, the agent iteratively selects key segments for video understanding, via training with gated bi-level reward system including trajectory-level rewards (answer correctness) with turn-level rewards (frame-query relevance).

**Compliance With Llm Reviewing Policy:**

Affirmed.

**Final Justification:**

The rebuttal addressed most concerns.

**Key Questions For Authors:**

Please check Originality in the Section of Strengths And Weaknesses.

**Limitations:**

yes

**Strengths And Weaknesses:**

* Soundness: RL method is a good direction for deeply understanding videos with complex contents. This work follows this direction via RL with a gated bi-level reward system.

* Presentation: This work is clearly written and well structured.

* Significance: RL is an important topic for video understanding community. However, the main idea of this work is basically similar to the existing approaches such as VideoChat-R1.5, VideoChat-A1, etc.

* Originality:

1) The originality of this work is limited. First, in terms of RL, VideoChat-R1.5 also reinforces multimodal reasoning by iterative perception. Please clarify the differences. Second, in terms of agentic frameworks, many existing agents (VideoAgent, VideoChat-A1, VCA, etc) leverage multi-turn reasoning styles for video understanding.  Hence, the design novelty of  this work is relatively limited.

2) The reward design is relatively straightforward. The trajectory-level reward is used to evaluate answer correctness, and the turn-level reward is used to evaluate frame-query relevance. Both are the basic design idea for long video understanding.

3) The performance comparsion should be further enhanced. Please compare the proposed method with VideoChat-R1.5 and Agentic Frameworks in the literature (VideoAgent, VideoChat-A1, VCA, etc).

---

> ### Author Rebuttal · Authors · 2026-03-31
>
> Dear Reviewer tZbB,
>
> We sincerely thank you for the thorough reviews and constructive feedback. We now present additional experiments and clarification regarding your concerns below.
>
> **W1: limited originality in terms of RL / “multi-turn”**
>
> We thank the reviewer for this important comment. We would like to clarify that the novelty of Video-MTR does **not** lie in simply combining RL with a multi-turn or agentic inference style. Rather, our contribution is a **goal-aligned multi-turn credit assignment mechanism** that makes long-video multi-turn reasoning trainable end-to-end under single-stage **pure RL, using only $\sim$8K examples and 1–2 training** epochs, without the usual SFT+RL pipeline or large-scale supervision.
>
> First, whereas VideoChat‑R1.5 focuses on test‑time scaling, ours centers on an efficiency‑oriented multi‑turn RL optimization design. VideoChat‑R1.5 uses a simple linear combination of spatio-temporal localization and final-answer rewards, and relies on the newly annotated **VTTS-80K**, a large-scale dataset , which, to our knowledge,**has not been publicly released**. In contrast, we introduce a goal‑aligned multi‑turn credit assignment with careful reward shaping, training on only  $\sim$8K examples curated from open‑source video grounding data. At inference time, this difference is also clear: VideoChat-R1.5 scales to up to 2048 frames during iterative perception, whereas Video-MTR is explicitly capped at **80** frames under a much smaller visual budget. Thus, our main claim is not higher raw-capacity scaling, but that carefully designed reward shaping can make multi-turn long-video policy learning practical and highly data-efficient.
> Second, compared with prior agentic video frameworks such as VideoAgent, VideoChat-A1, and VCA, our difference is also fundamental. Those methods mainly rely on training-free inference heuristics, external tools, or hand-designed multi-stage pipelines to improve exploration. They do not directly optimize the backbone MLLM and rely on external components . In contrast, Video-MTR turns multi-turn long-video reasoning into an **end-to-end trainable** capability through pure RL post-training, enabling a **unified model** to perform dynamic grounding and reasoning, eliminating the need for external modules.
>
> **W2:  Reward design is relatively straightforward**
>
> Please see our detailed response to R1-W1, “Novelty beyond a simple accuracy + tIoU reward.”
>
> **W3: Additional comparisons with recent work**
>
> We thank the reviewer for the suggestion. We now add a dedicated comparison with more representative agentic frameworks in Table A. We acknowledge that some recent systems report higher absolute scores on certain benchmarks; however, they operate under different assumptions and resource budgets. VCA and VideoAgent rely on closed-source frontier models such as **GPT-4o**. VideoChat-R1.5 is a visual test-time scaling method with a much larger dynamic inference budget, scaling up to **2048** frames. VideoChat-A1 is a heterogeneous chain-of-shot agent pipeline that relies on external LongCLIP-based modules and incurs substantially higher processing cost, since it performs 1 fps feature extraction over the full video and further processes candidate shots during selection and partition. In contrast, Video-MTR is an **efficient unified** framework trained and evaluated under strict resource budgets: it uses no external modules, and the entire system is capped at **80** frames. These comparisons highlight the strength of our cleaner, lighter-weight, and more data-efficient formulation.
>
> Table A. Comparisions with agentic methods.
>
> | Method | Avg. Frames | VideoMME (w/o sub.) | MLVU (test) | LongVideoBench | LVBench | EgoSchema |
> |--------|------------|---------------------|-------------|----------------|---------|-----------|
> | VideoAgent [1] | 87 | 56 | – | – | 29.3 | 60.2 |
> | VideoMemAgent [2] | 72 | 57.4 | – | – | – | 62.8 |
> | VCA | 20 | – | – | – | 41.3 | 73.6 |
> | VideoChat-A1  (Qwen2.5-VL-7B) | 42 | 71.8 | – | 64.2 | – | 70.7 |
> | VideoChat-R1.5-7B-M  | 4～2048 | 67.1 | – | 62.6 | 48.4 | – |
> | **Video-MTR (Ours)** | 80 | 62.7 | 50.4 | 57.1 | 42.3 | 68.8 |
>
> [1] Wang  et al., Videoagent: Long-form video understanding with large language model as agent
>
> [2] Fan et al., Videoagent: A memory-augmented multimodal agent for video understanding

---

> > ### Author Rebuttal · Reviewer_tZbB · 2026-04-03
> >
> > The rebuttal addressed most concerns.

---

> > > ### Author Response · Authors · 2026-04-06
> > >
> > > Thank you for the positive feedback and for updating your score. We greatly appreciate your support.

---

### Official Review · Reviewer_RxR3 · 2026-03-15

**Soundness:** 3
**Presentation:** 3
**Significance:** 3
**Originality:** 2
**Overall Recommendation:** 4
**Confidence:** 3

**Summary:**

The paper proposes Video-MTR, an RL–based multi-turn reasoning framework that treats video understanding as an interactive decision process. Instead of processing a fixed set of frames in a single pass, the model iteratively retrieves relevant video segments conditioned on prior observations and the query. The method is trained via pure RL post-training without supervised fine-tuning.

The core technical contribution is a gated bi-level reward system, including:
1. trajectory-level reward: final answer correctness
2. turn-level reward: relevance of retrieved frames
3. goal-gated shaping: intermediate rewards granted only when the final answer is correct

Overall, Video-MTR can achieve competitive performance on multiple long-video benchmarks using substantially less training data than prior approaches, while employing a multi-turn RL reasoning paradigm. Extensive experiments are conducted on VideoMME, MLVU, LongVideoBench, LVBench, and EgoSchema, showing consistent improvements over the backbone and favorable performance compared with several baselines under similar frame budgets.

**Compliance With Llm Reviewing Policy:**

Affirmed.

**Final Justification:**

The authors addressed all the concerns, explaining the reward design, frame sampling, and added more evaluations that support the conclusion. Thus, I am willing to adjust to weak accept.

**Key Questions For Authors:**

Please check the weakness part. I will consider updating the ratings if the questions can be well answered.

**Limitations:**

Yes

**Strengths And Weaknesses:**

Strength:
1. The bi-level reward design addresses a known challenge in multi-step RL
2. The experimental results demonstrate the effectiveness of Video-MTR
3. The paper is well-written.

Weaknesses:
1. The contribution of the paper mainly focuses on the reward design. However, the combined reward (acc + tiou) is straightforward and has been utilized by many related tasks, which hurts the novelty of the submission, since the multi-turn video reasoning has also been explored earlier, e.g.,  SVBench[1], SAMA[2].
2. In the main table, it shows the results of the framework with frame numbers 32, 64, or 64. In line 330, the authors also mentioned the results of 768 frames. However, the intermediate frame number is not reported, which is important to investigate the impact of frame number.
3. The evaluated benchmarks mainly focus on popular video understanding benchmarks; some reasoning-heavy benchmarks (e.g., MMVU[3], Video MMMU[4], Video MMLU[5], etc.) may also be considered to further prove the effectiveness of Video-MTR.

[1] Yang et al., SVBench: A Benchmark with Temporal Multi-Turn Dialogues for Streaming Video Understanding
[2] Sun et al., SAMA:Towards Multi-Turn Referential Grounded Video Chat with Large Language Models
[3] Zhao et al., MMVU: Measuring Expert-Level Multi-Discipline Video Understanding
[4] Hu et al., Video-MMMU: Evaluating Knowledge Acquisition from Multi-Discipline Professional Videos
[5] Song et al., Video-MMLU: A Massive Multi-Discipline Lecture Understanding Benchmark

---

> ### Author Rebuttal · Authors · 2026-03-31
>
> Dear Reviewer RxR3，
>
> We sincerely thank you for the careful reading and constructive feedback. We provide additional experiments and clarifications below.
>
> **W1: Novelty beyond a simple “accuracy + tIoU” reward.**
>
> We thank the reviewer for the thoughtful comment. We agree that simply combining answer correctness with a retrieval-quality signal has, by itself, been explored in this domain. Our core contribution is a **goal-aligned multi-turn credit assignment mechanism** that makes multi-turn long-video reasoning trainable in a highly **data-efficient** manner, eliminating the need for the commonly used two-stage SFT+RL pipeline and large-scale training data. Video-MTR is trained with single-stage pure RL using only $\sim$8K examples for 1–2 epochs.
>
> Specifically, in Video-MTR, the turn-level reward is **not a naive acc + tIoU** combination. It is defined on  marginal retrieval improvement, which explicitly penalizes non-progressive retrieval. More importantly, the reward is goal-gated by final answer correctness, which couples intermediate evidence gathering with end-task success and prevents the reward-hacking behavior as illustrated in Fig. 5. This design is further combined with token-level credit propagation across turns and exploration bootstrapping.  As a result, training remains stable from the beginning, and Video-MTR still matches or surpasses other post-training approaches trained with **257K to 4.4M** examples. The effectiveness of this design is also reflected in comparison with contemporaneous methods TimeSuite[1] and Temporal-RLT[2], both of which exploit temporal supervision (e.g., accuracy + tIoU): TimeSuite uses **432K** grounded instruction-tuning samples, and Temporal-RLT trains on **32K selected examples from a 490K** pool under a dual reward of answer accuracy and temporal-grounding tIoU.
>
> Finally, the cited prior works SVBench and  SAMA  study a different notion of “multi-turn” from ours. Their multi-turn is defined by the task setup: the model must handle multiple related questions within a dialogue; ours  is a property of the method, the model iteratively retrieves and reasons over multiple turns to deepen understanding of a single question. These are fundamentally different settings.
>
> [1] Zeng et al., TimeSuite: Improving MLLMs for Long Video Understanding via Grounded Tuning, ICLR 2025 .
>
> [2] Li et al., Reinforcement Learning Tuning for VideoLLMs: Reward Design and Data Efficiency (Temporal-RLT)
>
> **W2: Intermediate frame budgets and scalability.**
>
> We would like to clarify that the 768-frame result refers to the **officially supported maximum input** of the backbone model Qwen2.5-VL-7B, not to an RL-trained Video-MTR variant. We include this result  to show that Video-MTR(**80f**) can **approach, or even surpass**, a much larger-input backbone while operating under a substantially smaller and more practical frame budget.
> Due to the computational cost of multi-turn RL, our training is currently capped at 80 frames. This budget is also consistent with current long-video post-training practice: most methods operate at ($\leq$ 128) frames, while only a few specialized long-context approaches (e.g., Video-XL, LongVA) scale beyond this range. In addition, we already report results at 32, 64, and 80 frames and analyze the scaling trend. Performance improves consistently as the frame budget increases, demonstrating clear scalability within the feasible budget range.
>
> **W3: Evaluation on reasoning-heavy benchmarks**
>
> Following standard long‑video understanding practices, we report results on the five mainstream LVU benchmarks for fair comparison. In the rebuttal, following the reviewer’s suggestion, we further add results on the more reasoning-heavy Video-MMMU benchmark. Notably, despite using a much smaller frame budget, **Video-MTR(80f) achieves 47.10%** overall accuracy, which is already comparable to **Qwen2.5-VL-7B(768f) at 47.44%**.
> We further analyze the three stages of Video-MMMU: Perception, Comprehension, and Adaptation. The official Qwen2.5-VL-7B(768f) result is 58.33%/ 44.33%/ 39.67%, while Video-MTR(80f) achieves 65.67% / 39.33%/ 36.30%. This pattern is consistent with the design of our method. Perception is **well aligned with long-video understanding**, as it depends heavily on locating fine-grained, question-relevant evidence over long temporal contexts; Video-MTR’s iterative retrieval directly strengthens this ability, yielding a substantial gain (+7.34%.). By contrast, Comprehension and Adaptation rely more on broader reasoning and general knowledge.  Since our post‑training does not include reasoning‑heavy data, our task‑specific training inevitably incurs some loss in generality. Overall, this experiment further supports that Video-MTR is particularly effective for **detail-sensitive long-video reasoning** under limited frame budgets.

---

> > ### Author Rebuttal · Reviewer_RxR3 · 2026-04-03
> >
> > Thanks for the detailed response. I will adjust my score to weak accept.

---

> > > ### Author Response · Authors · 2026-04-06
> > >
> > > We are pleased that our clarifications have addressed your concerns, and we greatly appreciate your support and the updated assessment.

---

### Decision · Program_Chairs · 2026-04-30

**Decision:**

Accept (regular)

**Comment:**

This paper receives: weak accept, weak accept, weak accept, weak accept. The reviewers all appreciate the contribution of this paper by focusing on dynamically selecting the frames and performing multi-turn reasoning with new reward design. The new techniques are verified on the task of long video understanding. In general, the AC agrees with reviewers and makes an accept recommendation. Meanwhile, the AC also agrees with one reviewer on the concern of scalability issue of the method. The authors also miss a related work on long video understanding with high performance (VideoChat-Flash: Hierarchical Compression for Long-Context Video Modeling). The authors are encouraged to include these discussions in the final version.